# Identification of Biomarkers for Methamphetamine Exposure Time Prediction in Mice Using Metabolomics and Machine Learning Approaches

**DOI:** 10.3390/metabo12121250

**Published:** 2022-12-10

**Authors:** Wei Sheng, Runbin Sun, Ran Zhang, Peng Xu, Youmei Wang, Hui Xu, Jiye Aa, Guangji Wang, Yuan Xie

**Affiliations:** 1China Pharmaceutical University Nanjing Drum Tower Hospital, Nanjing 210000, China; 2Key Laboratory of Drug Metabolism and Pharmacokinetics, State Key Laboratory of Natural Medicines, China Pharmaceutical University, Nanjing 210009, China; 3China National Narcotics Control Commission—China Pharmaceutical University Joint Laboratory on Key Technologies of Narcotics Control, China Pharmaceutical University, Nanjing 210009, China

**Keywords:** biomarker, drug abuse, methamphetamine, metabolomics, machine learning

## Abstract

Methamphetamine (METH) abuse has become a global public health and safety problem. More information is needed to identify the time of drug abuse. In this study, methamphetamine was administered to male C57BL/6J mice with increasing doses from 5 to 30 mg kg^−1^ (once a day, i.p.) for 20 days. Serum and urine samples were collected for metabolomics studies using gas chromatography–mass spectrometry (GC-MS). Six machine learning models were used to infer the time of drug abuse and the best model was selected to predict administration time preliminarily. The metabolic changes caused by methamphetamine were explored. As results, the metabolic patterns of methamphetamine exposure mice were quite different from the control group and changed over time. Specifically, serum metabolomics showed enhanced amino acid metabolism and increased fatty acid consumption, while urine metabolomics showed slowed metabolism of the tricarboxylic acid (TCA) cycle, increased organic acid excretion, and abnormal purine metabolism. Phenylalanine in serum and glutamine in urine increased, while palmitic acid, 5-HT, and monopalmitin in serum and gamma-aminobutyric acid in urine decreased significantly. Among the six machine learning models, the random forest model was the best to predict the exposure time (serum: MAE = 1.482, RMSE = 1.69, R squared = 0.981; urine: MAE = 2.369, RMSE = 1.926, R squared = 0.946). The potential biomarker set containing four metabolites in the serum (palmitic acid, 5-hydroxytryptamine, monopalmitin, and phenylalanine) facilitated the identification of methamphetamine exposure. The random forest model helped predict the methamphetamine exposure time based on these potential biomarkers.

## 1. Introduction

Methamphetamine (METH) is a common addictive drug abused extensively world- wide. Thirty-five million people took methamphetamine and 585,000 died in 2017 worldwide, according to the World Drug Report 2019. About 284 million people in the world have used drugs in the past 12 months, including 34 million amphetamine users according to the World Drug Report 2021. Methamphetamine [1] is a powerful central nervous stimulant that causes persistent damage to nerve terminals and influences attention, heart rate, blood pressure, body temperature, and appetite in the short term. Long-term use or high doses of METH usually lead to psychosis, depression, paranoia, severe anxiety, and violence. Many studies focus on changes in neurotransmitters in the central nervous system affected by methamphetamine, such as dopamine, 5-hydroxytryptamine (5-HT), noradrenaline, and adrenaline [2], and the following effect caused by methamphetamine, such as oxidative stress, neuroexcitatory toxicity [3], and neuroinflammation [4]. In addition, some studies revealed that neuropeptide Y and oxytocin were affected by METH [5,6]. The complete metabolic changes caused by METH deserve to be paid attention to.

Metabolomics [7,8] was developed in the mid-1990s as a new discipline for qualitative and quantitative analysis of small molecular metabolites in an organism or cell. Metabolomics can quickly and intuitively reflect the metabolic changes in the body, discover biomarkers that can be used to predict disease, and reveal the mechanism of change in the body under the action of drugs or diseases. It has been used to study various metabolic diseases such as cancer, hepatitis, hyperlipidemia, diabetes, etc. [9,10,11,12]. Many studies have used non-targeted and targeted metabolomics to reveal the metabolic effects associated with the short-term withdrawal of methamphetamine [13,14,15,16,17,18]. Metabolites in rat plasma were collected 16 days after methamphetamine self-doping and 12 or 24 h after withdrawal, and rapid and significant differences were found in metabolic pathways involving energy metabolism, the nervous system, and membrane lipid metabolism [14]. Similarly, after METH injections in rats for five consecutive days and withdrawal for two days, it was found that branch chain amino acids were significantly consumed, and the TCA cycle and lipid metabolism accelerated. Creatinine, citrate, and 2-ketoglutaric acid in serum and lactic acid in urine were selected as biomarkers [13]. Another study recruited 80 methamphetamine abusers and 80 healthy individuals, and metabolomics study of serum from the two groups revealed 16 differential metabolites (cholic acid, deoxycholic acid, hydroxylamine, etc.). Three potential abuse markers (RNA, cellobiose, and malt sugar) were identified and the area under the receiver operator characteristic curve was 0.975 [17].

Drug use affects the nervous system. Methamphetamine injection causes neurotransmitter disturbance, such as tryptophan and 2-hydroxyglutaric acid, and imbalance between oxidative stress and antioxidants and gliosis in the hippocampus, nucleus accumbens (NAc), and prefrontal cortex (PFC) [15,16].These studies focused on short-term drug use. They did not reveal the relationship between the time of drug abuse and metabolic change. Time may be a reliable and objective indicator to describe drug abuse. The description of drug use still lacks objective and dependable conditions. The brain often represents and predicts the addiction degree, such as the resting-state functional magnetic resonance imaging data and cerebral blood flow [19]. These methods have limitations on personnel willingness, location, and instrument. In our previous studies [20], metabolomics could reflect changes in the metabolic state of metabolic disease in different periods, including morphine, heroin [21,22,23], and methamphetamine [13], which altered metabolic status during periods of exposure, withdrawal, and relapse in serum or urine. These studies suggest that metabolomics has the ability to reflect changes in the metabolic rate at different times of drug use through changes in the peripheral circulation.

In this study, we aimed to reveal the metabolomic pattern of serum and urine of mice who were exposed to methamphetamine at different times. By analyzing the metabolomics data, we tried to determine the relationship between potential biomarkers and drug exposure time based on the selected algorithm to predict drug exposure time accurately.

## 2. Experimental Design

### 2.1. Regents

Methamphetamine was supplied by Drug Intelligence and Forensic Center, Ministry of Public Security (Beijing, China). Myristic-1,2-^13^C_2_ acid (99 atom%^13^C), methoxamine hydrochloride, pyridine, N-Methyl-N-(trimethylsilyl)trifluoroacetamide (MSTFA) with 1% trimethylchlorosilane (TMCS), methanol (chromatographic purity), and n-heptane (chromatographic purity) were purchased from Sigma-Aldrich (St. Louis, MO, USA). Purified water was produced using a Milli-Q system (Millipore, Bedford, MA, USA).

### 2.2. Instruments

The equipment included a Sorvall Biofuge Stratos centrifuge (Thermo Fisher Scientific, MA, USA), vacuum rotary drying apparatus (Thermo Fisher Scientific, MA, USA), and Shimadzu GCMS-QP2010 (Shimadzu Corp., Tokyo, Japan). The gas chromatograph was equipped with a capillary column with silanization fused-silica capillary tube (DB-5 fixed phase, thickness 0.18 m, 10 m × 0.18 mm diameter, J&W Scientific, CA, USA).

### 2.3. Animal and Sample Preparation

Male C57BL/6J mice (20–24 g) were purchased from Sippr/bk Laboratory Animal Co. Ltd. (Shanghai, China). All animal experimental procedures met the guidelines of the Association for Assessment and Accreditation of Laboratory Animal Care and the Institutional Animal Care and Use Committee of China Pharmaceutical University. After a one-week adaptation, the mice were divided into six groups: the normal control group, 10-day group, 15-day group, 20-day group, withdrawal group, and relapse group according to the exposure time. Methamphetamine was given once a day through intraperitoneal injections at a dose range from 5 mg/kg to 30 mg/kg (specific administration program shown in Figure 1). After withdrawal, the dose of 22 mg/kg was given continuously for five days to the relapse group. The serum samples were taken from the eye socket veins and urine samples were collected in metabolic cages. All samples were stored at −80 °C.

### 2.4. Sample Preparation for GC/MS Analysis and Compound Identification

Samples were prepared according to the established protocol in [13,22]. Briefly, methanol solution with internal standard compound (myristic-1,2-^13^C_2_ acid, 2.5 μg/mL) was added to the serum or urine (urine was added to 10 mg/mL urea enzyme solution at 37 °C for 1 h to break down urea) to precipitate protein. Then, the solution was mixed at 4 °C for 1 h and centrifugated. The supernatant was transferred into a sample bottle and vacuum concentrated to dry. Then, methoxyamine in pyridine (10 mg/mL) was added for oximation and a trimethylsilanization reagent was added for silanization. Finally, the external standard methyl myristate in heptane (30 μg/mL) was added. The mixed solution was analyzed using gas chromatography/mass spectrometry (GC/MS). Samples were separated on a 10 m quartz capillary column (DB-5), which used helium as the carrier gas. The gasification temperature was 250 °C with a 30 °C/min rate of warming. The temperature increased from 70 to 310 °C. Then, the sample was scanned by 30 V/s in the MS. The tuning bombardment electron flow energy was 70 eV to obtain a total ion flow map over a 50 to 800 mass-to-load ratio range. Samples were analyzed in a random order. After the analysis, compounds were identified by comparing ion characteristics, such as m/z, retention time, and referring to the standard database: the National Institute of Standards and Technology (NIST) library 14; Wiley 9 (Wiley-VCH Verlag GmbH & Co. KGaA, Weinheim, Germany); the inhouse mass spectral library database established by Umeå Plant Science Center (Umeå University, Sweden); and the Key Laboratory of Drug Metabolism and Pharmacokinetics, China Pharmaceutical University (Nanjing, China).

### 2.5. Machine Learning Methods for the Prediction of Drug Exposure Time

Six machine learning methods, including partial least square (PLS), principal component regression (PCR), support vector machine (SVM), random forest (RF), K-nearest neighbor (KNN), and Bayesian-regularization neural networks (BRNNs) were used for regression. We used variable importance in the projection (VIP) analysis to find the metabolites contributing to the regression in each model. Mean absolute error (MAE), root mean squared error (RMSE), and R squared (R^2^) were used to evaluate models. All the machine learning methods were performed in the R Project (version 3.6.3) and tuned with the “caret” package.

### 2.6. Statistical Analysis

The peak area was corrected by the internal standard (to eliminate the interference of urine volume and creatinine correction of urine) and then imported into SIMCA-P 13 software (Umetrics, Umeå, Sweden) for multistatistical analysis, including principal component analysis (PCA) and partial least square multiplication analysis (PLS-DA). Statistical analysis used one-factor variance analysis (ANOVA), with a significance level of 0.05, followed by post hoc pairwise comparisons adjusted by the Benjamini–Hochberg method performed by the R Project (version 3.6.3) to control the false discovery rate (FDR).

## 3. Results

### 3.1. Endogenous Metabolite Identification

The typical chromatograms of serum and urine are shown in Figure 2. By comparing the measured spectrum with the spectrum of the reference compound in the metabolite databases, we identified 83 compounds in serum and 72 compounds in urine, including fatty acid metabolism, glucose metabolism, and TCA cycle-related compounds (Appendix A).

### 3.2. Multivariate and Univariate Data Analysis

To compare the differences between each group of serum and urine, we imported the data into SIMCA 13.0 and used the PLS-DA model for comparative analysis. As shown in Figure 3, spots gathered within the same dose group. The groups followed a dose-dependent trend and there were significant differences between each methamphetamine group and the control group. In both serum and urine, the metabolic patterns of the 10-day, 15-day, and 20-day groups gradually deviated from the control group, the metabolic patterns of the 15-day withdrawal group returned to be close to the 10-day group, and the metabolic patterns of the 20-day relapse groups were close to the 20-day group. This result indicated that methamphetamine would seriously affect the metabolic patterns of mice and gradually change them over time, while withdrawal can restore the metabolic patterns to a certain extent, but not fully return them to the normal level. In addition, the most apparent difference between the 15-day group and the control group in urine and the metabolic patterns of the 20-day group present a trend of shifting toward the control group (Figure 3b). These results suggested tolerance may have occurred in the 20-day group. After withdrawal, metabolic patterns quickly returned to those of the 10-day group, and resorption metabolic patterns were close to the control group, indicating that the metabolic change in urine after methamphetamine administration can easily affect metabolic patterns. Similarly, metabolic pattern quickly returned to the control level after withdrawal.

### 3.3. Machine Learning Method Selection and Optimization

Six machine learning models were used to process serum metabolomics data for 0, 10, 15, and 20 days, including RF, PCR, PLS, SVM, BRNN, and KNN. By comprehensively comparing the regression evaluation indexes MAE, RMSE, R^2^ (Figure 4a, Appendix A), and the verification results of the test set (Appendix A), the random forest model was selected for serum samples with high accuracy of regression (MAE = 1.482, RMSE = 1.69, R^2^ = 0.981) (Figure 4b) and the accuracy of verification results is also high for 20% of randomly sampled test sets (MAE = 1.922, RMSE = 2.528, R^2^ = 0.934) (Figure 4c). Similarly, the random forest model was also selected for urine samples, with high regression accuracy (MAE = 2.369, RMSE = 1.926, R^2^ = 0.946) (Figure 4e), and the prediction accuracy of 20% random sampling was also good (MAE = 5.094, RMSE = 4.504, R^2^ = 0.952) (Figure 4f). SVM and BRNN performed well in the training set (Appendix A), but poorly in the test set verification (Appendix A).

### 3.4. Metabolic Pathway Analysis

The top 20 compounds in the variable importance analysis in the random forest model of serum and urine were selected as differential compounds (Figure 5a,b). Through the enrichment of metabolic pathways and concrete analysis, we found that methamphetamine mainly influenced glutamate and glutamine metabolism, alanine, aspartate, and glutamate metabolism, and glyoxylic acid and dicarboxylic acid metabolism both in the serum and urine. Glycerolipid metabolism and biosynthesis of unsaturated fatty acids were changed in the serum, and glycine, serine, and threonine metabolism, glutathione metabolism, and purine metabolism were altered in the urine (Figure 5e,f).

By observing the specific changes in different compounds in the serum and urine of mice (Figure 5a,b, Table 1), it was found that the continuous administration of methamphetamine dramatically affects the energy metabolism in mice. The serum primarily manifests enhanced amino acid metabolism and increased fatty acid consumption. In contrast, urine mainly displayed a slower TCA cycle, increased organic acid excretion, and abnormal purine metabolism.

The overall level of fatty acids in serum decreased. Lauric acid, myristic acid, oleic acid, palmitic acid, stearic acid, octadecanoic acid, cis-9-hexadeceroenoic acid, cis-9-hexadecenotanoic acid, arachidonic acid, 1-monostearin, monopalmitin, and cis-4,7,10,13,16,19-docosahexaenoic acid all decreased significantly after METH administration, increased dramatically after short-term withdrawal, and returned to levels similar to those of the 20-day group with continuous administration after relapse. In addition, the cholesterol increased significantly, and the glycerol decreased significantly. This result indicates that the energy consumption increases after methamphetamine administration, the energy demand of the body increases, and the fatty acids are rapidly consumed (Table 1). For carbohydrate metabolism, pyruvate, lactate, glycerol-3-phosphate, and fructose are all significantly reduced. However, the metabolism of amino acids in serum showed an upward trend. Phenylalanine, methionine, alanine, proline, serine, valine, leucine, isoleucine, glutamine, and aspartate all increased significantly, and glutamate, aspartate, and serotonin decreased significantly. These results indicate that the body’s glucose and lipid substances may be consumed in large quantities, and amino acids may begin to be consumed for energy supply. At the same time, in the TCA cycle, fumarate, malate, citrate, and succinate decreased, while α-ketoglutaric acid increased significantly. The relapse group showed a more significant increase than the continuous administration for 20 days group, and the TCA cycle was also greatly affected. In general, many metabolites in the serum underwent substantial changes after administering methamphetamine. The fatty acids and glucose metabolism-related substances tended to recover after withdrawal. In contrast, metabolites in the TCA cycle and amino acid metabolism struggled to recover to initial levels in the short term and showed more pronounced changes after relapse, such as phenylalanine, methionine, glutamine, and asparagine (Table 1).

Organic acid excretion in the urine increased, such as oxalic acid and glutaric acid. 3- hydroxybutyric acid increased significantly after relapse. Purine metabolism including hypoxanthine, uric acid, and guanine increased significantly, and ornithine decreased substantially in the urea cycle. The TCA cycle also showed an overall downward trend, and glutamine showed a significant increase similar to serum. However, α-ketoglutaric acid, unlike in serum, shows a downward trend in the urine (Table 1).

### 3.5. Biomarker Selection and Random Forest Model Optimization to Predict Drug Exposure Time

Correlation analysis was performed between the time of drug exposure and metabolite concentrations. In the serum, palmitic acid, 5-hydroxytryptamine, monopalmitin, phenylalanine, glycerol-3-phosphate, and oleic acid had the most significant correlation coefficient (Appendix A). In urine, glutamine, oxalic acid, hypoxanthine, guanine, glutaric acid, uric acid, and gamma-aminobutyric acid had the most significant correlation coefficient (Appendix A). We tried to predict drug exposure time using the random forest model with different groups of metabolites which are shown in Appendix A. The group of palmitic acid, 5-hydroxytryptamine, mono-palmitin, and phenylalanine in serum was found to have the largest R squared, and smallest RAE and RMSE (Appendix A). These four metabolites can be considered as biomarkers of methamphetamine exposure in serum (Figure 6a). In urine, the group of glutaric acid, guanine, hypoxanthine, oxalic acid, glutamine, uric acid, and gamma-ami nobutyric acid was found to have the largest R squared, and smallest RAE and RMSE (Appendix A). These metabolites can be considered biomarkers of methamphetamine exposure in the urine (Figure 6b). According to the results above, the RF model established with metabolites in serum is better than the one with those in urine and is more suitable for predicting the time of drug use.

## 4. Discussion

By studying the metabolomics of mice with different administration times of meth- amphetamine, this study found that the metabolic patterns of mice with varying durations of administration gradually shifted from the control group, which in the serum was more apparent. The urine results were reversed after 20 days of administration, which may because the body tolerated it after a long period and high dose intake. In addition, withdrawal and relapse also significantly influence the metabolic pattern. In both serum and urine, the metabolic patterns of the withdrawal group are close to those of the 10-day and control groups. In contrast, the metabolic patterns of the relapse group are close to that of the 15-day and 20-day dosing groups. This result suggests that variation of metabolic patterns depends on the dosage and duration of drug exposure and drug withdrawal and relapse. In addition, other metabolic-related factors, such as diseases, might also influence the metabolic pattern of dependent users of drugs.

Therefore, with the help of the caret package of the R Project, six machine learning models, including RF, PCR, PLS, SVM, BRNN, and KNN, were used to predict drug use duration with mouse serum and urine metabolomics data. After comprehensive comparison, the random forest method with good training results, strong generalization ability, and short running time was selected as the best model. The random forest model [24] is an integrated learning method based on the decision tree, which generates multiple trees by random sampling and random sampling feature sets, and summarizes the resulting output of each tree. It is not easy to produce an overfitting phenomenon, it is suitable for data sets with missing data and non-equilibrium, and the established model is stable and the prediction accuracy is high. It can be applied to regression and classification problems at the same time, which is one of the best algorithms at present [25].

Metabolic profiling can identify potential biomarkers for drug addiction, withdrawal, and relapse since drug usage can drastically change metabolism. According to reports, taking heroin caused an increase in energy metabolism and a shift in serotonin levels [21]. Similarly, another study reported accelerated tricarboxylic acid activity, increased fatty acid β-oxidation, as well as a noticeable reduction in branched-chain amino acids during methamphetamine administration [13]. For the analysis of the specific metabolites and pathways, we found that the continuous administration of methamphetamine dramatically affects the energy metabolism including enhanced amino acid metabolism and increased fatty acid consumption in serum and disturbed TCA cycle, increased organic acid excretion, and abnormal purine metabolism in urine. Amino acid metabolism disorders may be due to dysfunction of liver metabolic enzymes [26]. Amino acids are catabolized to produce large amounts of ammonia. Glutamate is converted into glutamine by glutamine synthase (GS), thereby fixing ammonia, leaving the brain, and entering the bloodstream [27]. Therefore, the glutamate in the serum is significantly reduced, and the glutamine is increased considerably. Hyperammonemia caused by methamphetamine has been reported [28]. Phenylalanine, methionine, asparagine, and glutamine in serum increase significantly. Phenylalanine, methionine, and asparagine are nitrogen donors, while the rise of glutamine is a manifestation of enhanced ammonia clearance. In addition, ornithine in the urine is also significantly reduced, and the urea cycle clearance pathway of ammonia [29] also runs rapidly, excreting a large amount of urea in urine, which proves the increase in ammonia in mice after multiple administrations of methamphetamine. Since ammonia can only be cleared by glutamine synthase in the brain, and the process is energy-dependent [30] and needs ATP, the process of the glutamate–glutamine cycle excreting excess ammonia produces energy requirements [31] for astrocytes in the brain, affecting the cellular energy state and energy metabolism. α-Ketoglutaric acid is a precursor to the neurotransmitter glutamate and is the ultimate metabolite of many amino acids [27]. A significant increase in α-ketoglutarate and a substantial decrease in glutamate indicate that the TCA cycling process of neurons is also affected. However, GS in the brain gradually loses its activity during long-term ammonia clearance. Long-term use of methamphetamine increases ammonia in the brain, which may destroy the brain’s blood–brain barrier, resulting in oxidative damage and inflammation, and may lead to hepatic encephalopathy [32]. Methamphetamine is a powerful CNS stimulant, but at high dosages, it is cytotoxic to human midbrain dopaminergic neurons. A dopamine overflow in the striatum brought on by methamphetamine can result in an excessive glutamate release into the brain [33]. After administering methamphetamine, we noticed considerable changes in serotonin, glutamate, glutamine, and gamma-aminobutyric acid. These findings were consistent with the hypothesis of the metabolic influence of exposure in addiction and withdrawal [34]. Palmitic acid, 5-hydroxytryptamine, monopalmitin, and phenylalanine in serum can be used as potential biomarkers of methamphetamine exposure.

Both oxalic acid and glutaric acid showed a significant increase in the urine. Oxalic acid is produced through the metabolism of glyoxylic acid or ascorbic acid in the body and is excreted in the urine. Endogenous oxalic acid synthesis [35] occurs mainly in hepatocytes and glyoxylic acid is the primary precursor. Typically, most glyoxylic acid is converted to glycine or glycolic acid, while excess glyoxylic acid is metabolized to oxalic acid. Urine with high oxalic acid may cause kidney stones [36] and inflammation [37], leading to renal failure in extreme cases. A significant increase in oxalic acid may be associated with acute kidney injury due to methamphetamine abuse [38,39]. A substantial increase in glutaric acid may be due to the obstruction of lysine metabolism and tryptophan due to the abnormality of glutaryl-CoA dehydrogenase, the metabolite of which accumulates in large quantities in the body and is excreted by the urine [40]. The accumulation of glutaric acid is neurotoxic and can also lead to oxidative stress [41,42]. It can disturb glutamine degradation by inhibiting glutamate dehydrogenase, blocking metabolic associations between neurons and astrocytes, ultimately leading to astrocyte death [43]. A significant rise in hypoxanthine in the urine may also be related to oxidative stress in the body. It was metabolized to uric acid, which is a strong oxidizing agent [44]. Oxidative stress caused by methamphetamine leads to the compensatory rising of antioxidant substances. Oxalic acid, glutaric acid, glutamine, guanine, uric acid, gamma-aminobutyric acid, and hypoxanthine in urine can be used as potential biomarkers of methamphetamine exposure.

## 5. Conclusions

In this study, the serum and urine metabolomics data of C57BL/6J mice showed significant change after methamphetamine administration, withdrawal, and relapse. Differential metabolites were used as the potential biomarkers of methamphetamine exposure and predicted drug exposure time analysis. Six machine learning methods were established and compared. The random forest model was chosen as the best model to predict the methamphetamine exposure time based on the serum and urine metabolomics data. The model established by metabolites in the serum outperformed that in urine. The finally selected potential biomarkers are palmitic acid, 5-hydroxytryptamine, monopalmitin, and phenylalanine in serum with the highest accuracy.

## Figures and Tables

**Figure 1 metabolites-12-01250-f001:**
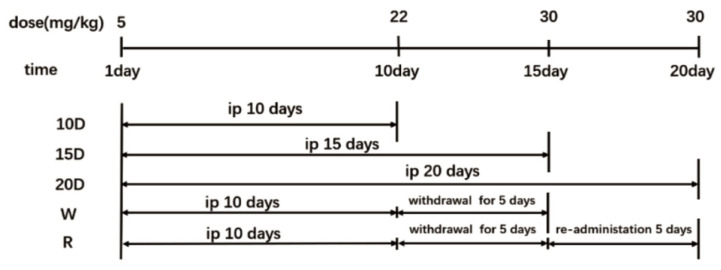
The scheme of drug administration. There are 5 groups in the experiment: 10-day group: given methamphetamine once a day through intraperitoneal injection at a dose range from 5 mg/kg to 22 mg/kg for 10 days; 15-day group: given methamphetamine once a day through intraperitoneal injection at a dose range from 5 mg/kg to 30 mg/kg for 15 days; 20-day group: given methamphetamine once a day through intraperitoneal injection at a dose range from 5 mg/kg to 30 mg/kg for 20 days; 15-day withdrawal group: given methamphetamine once a day through intraperitoneal injection at a dose range from 5 mg/kg to 22 mg/kg for 10 days, then methamphetamine administration stopped; 20-day relapse group: re-administration 5 days after the withdrawal.

**Figure 2 metabolites-12-01250-f002:**
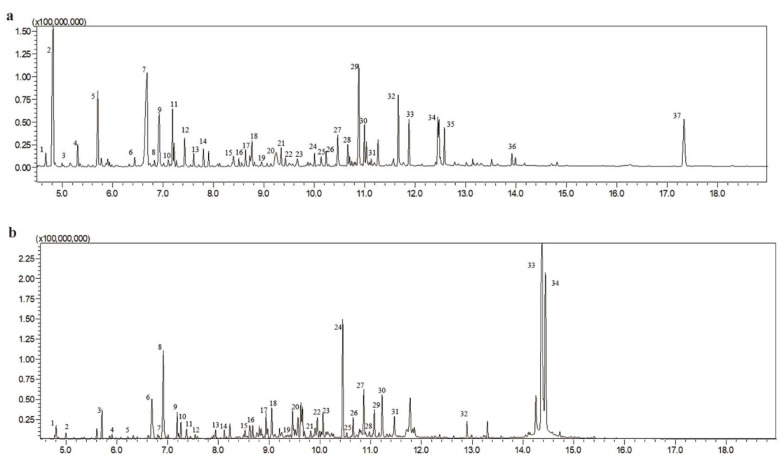
Representative total ion current chromatograms in serum and urine. (**a**) Typical com- pounds in serum as follows: 1. Pyruvate; 2. Lactate; 3. Glycolic acid; 4. Alanine; 5. Oxalic acid; 6. Valine; 7. Urea-2TMS; 8. Serine-2TMS; 9. Phosphate; 10. Proline; 11. Glycine-2TMS-2; 12. Uracil; 13. Serine-3TMS; 14. Threonine; 15. Aminomalonic acid; 16. Malate-3TMS; 17. Aspartate-3TMS; 18. Pyroglutamate-TMS; 19. Cysteine; 20. Glutamine-4TMS; 21. Glutamate-3TMS; 22. Phenylalanine; 23. Asparagine; 24. ES; 25. Glycerol-3-phosphate; 26. Glutamine-4TMS; 27. Citrate; 28. IS; 29. Ribitol; 30. Glucose; 31. Cis-9-hexadecenoic acid; 32. Palmitic acid; 33. Myo-Inositol; 34. Linoleic acid; 35. Octadecanoic acid; 36. Monopalmitin; 37. Cholesterol-TMS. (**b**) Typical compounds in urine: 1. Lactate; 2. Glycolic acid; 3. Oxalic acid; 4. 3-Hydroxybutyric acid; 5. Valine; 6. Urea-2TMS; 7. Serine-2TMS; 8. Phosphate; 9. Proline; 10. Succinate; 11. Glyceric acid-3TMS; 12. Fumarate; 13. Threonine; 14. b-Alanine-3TMS; 15. Glycine; 16. Pyroglutamate-TMS; 17. Cysteine; 18. 2-Ketoglutaric acid; 19. Glutamate-3TMS; 20. N-Acetylaspartate-2TMS; 21. Lyxose; 22. ES; 23. Aconitic acid; 24. Citrate; 25. Hypoxanthine; 26. IS; 27. Ribitol; 28. Glucose; 29. Lysine; 30. Glucitol; 31. Cys-Gly; 32. Cystine; 34. Lactose.

**Figure 3 metabolites-12-01250-f003:**
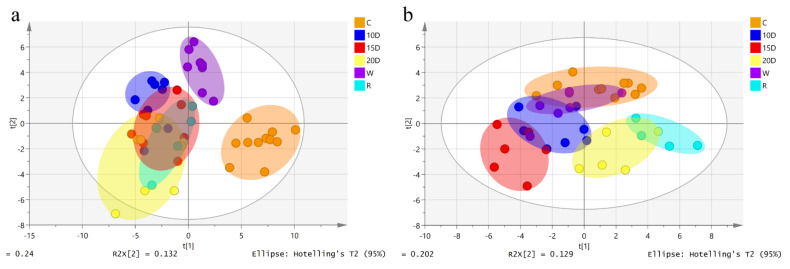
PLS-DA model of metabolic patterns for mice. (**a**) Serum. (**b**) Urine. The blue points represent the group of 10 days; the red points represent the group of 15 days; the yellow points represent the group of 20 days; the purple points represent the group of withdrawal (withdrawal for 5 days after 10 days’ administration); the cyan points represent the group of relapse (re-administration for 5 days after the withdrawal).

**Figure 4 metabolites-12-01250-f004:**
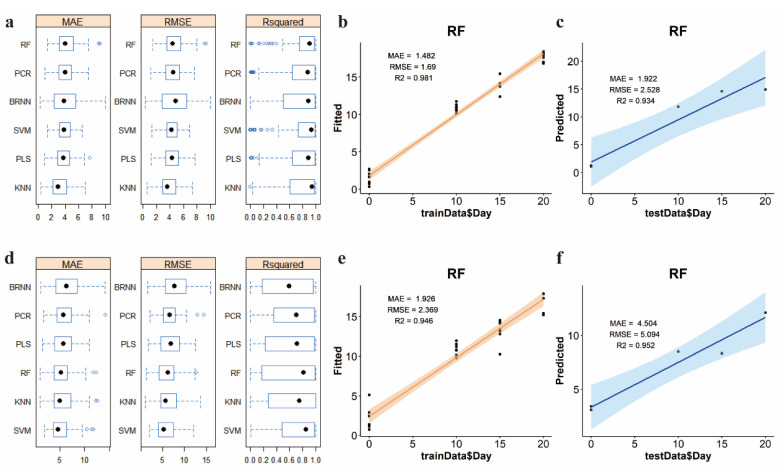
The training and testing results of six machine learning models. (**a**) The box diagram of training results of 6 machine learning models of serum. (**b**) The training results of the RF model of serum. (**c**) The time prediction results of the RF model of serum in the testing set. (**d**) The box dia- gram of training results of 6 machine learning models of urine. (**e**) The training results of the RF model of urine. (**f**) The time prediction results of the RF model of urine in the testing set.

**Figure 5 metabolites-12-01250-f005:**
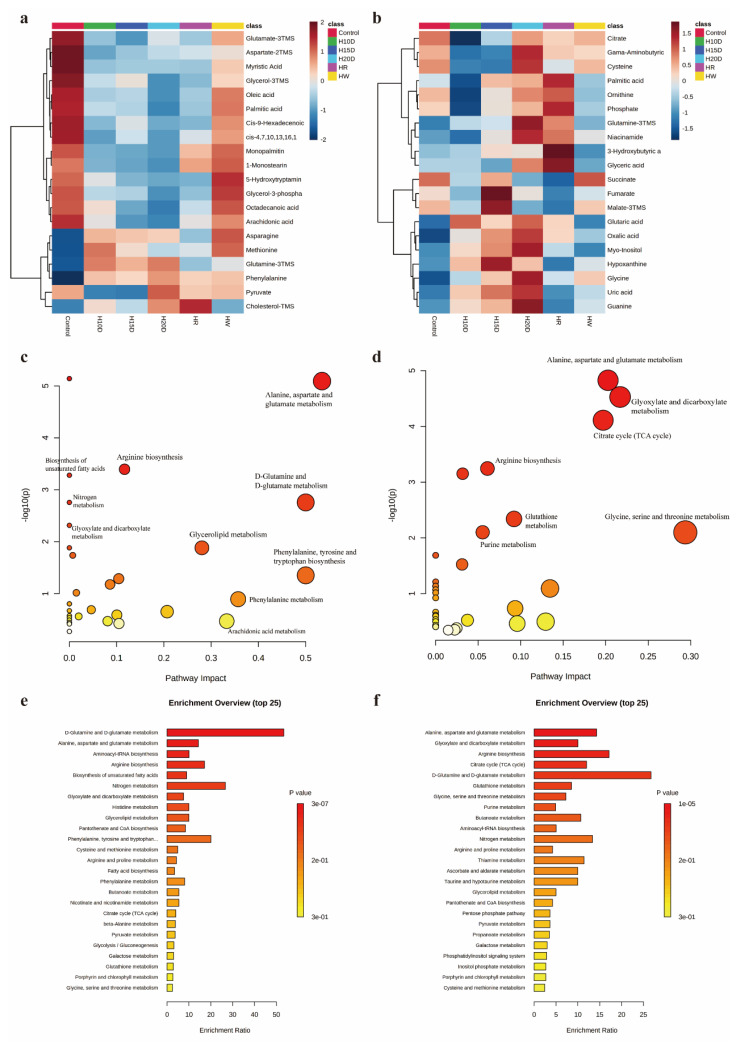
The heatmap of representative compound changes in serum and urine of methamphetamine mice and the metabolic pathways. (**a**) The heatmap of representative compound changes in serum. (**b**) The heatmap of representative compound changes in the urine. (**c**) The pathway impact of serum metabolites. (**d**) The pathway impact of urine metabolites. (**e**) Enrichment analysis of pathway-associated metabolite sets in serum. (**f**) Enrichment analysis of pathway-associated metabolite sets in urine.

**Figure 6 metabolites-12-01250-f006:**
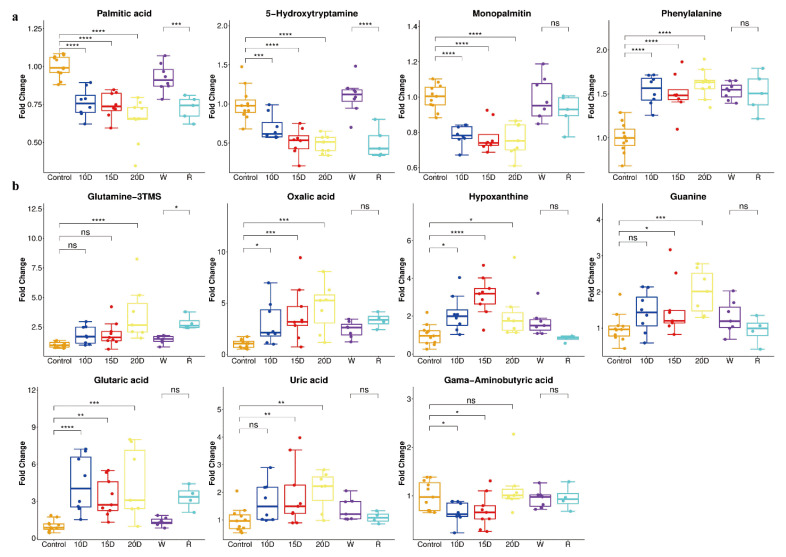
The biomarkers of serum and urine after METH administration. (**a**) The biomarkers of serum after METH administration. (**b**) The biomarkers of urine after METH administration. ns: *p* > 0.05, *: *p* < 0.05, **: *p* < 0.01, ***: *p* < 0.001, ****: *p* < 0.0001.

**Table 1 metabolites-12-01250-t001:** Differential metabolites in serum and urine.

Metabolic Pathways	Compounds	Serum					Urine				
		Methamphetamine Treatment			Withdrawal	Relapse	Methamphetamine Treatment			Withdrawal	Relapse
		10DvsC	15DvsC	20DvsC	Wvs15D	Rvs20D	10DvsC	15DvsC	20DvsC	Wvs15D	Rvs20D
Lipid metabolism	Lauric acid	↓ *	↓	↓ *	↑ *	-	—	—	—	—	—
	Myristic acid	↓ *	↓ *	↓ *	↑ *	-	—	—	—	—	—
	Oleic acid	↓ *	↓ *	↓ *	↑ *	-	—	—	—	—	—
	Palmitic acid	↓ *	↓ *	↓ *	↑ *	-	↓	-	↑	-	-
	Octadecanoic acid	↓	↓ *	↓ *	↑ *	↑	↓ *	↓	↓	↑	↑ *
	Cis-9-hexadecenoic acid	↓ *	↓ *	↓ *	↑ *	-	—	—	—	—	—
	Arachidonic acid	↓	↓ *	↓ *	↑ *	↑	-	↑	↑	-	-
	Monopalmitin	↓ *	↓ *	↓ *	↑ *	↑ *	—	—	—	—	—
	1-Monostearin	↓ *	↓ *	↓ *	↑ *	↑ *	—	—	—	—	—
	Cholesterol-TMS	↑	↑	↑ *	↓	↑	—	—	—	—	—
	Glycerol-3TMS	↓ *	↓ *	↓ *	-	-	—	—	—	—	—
	cis-4,7,10,13,16,19-docosahexaenoic acid	↓ *	↓ *	↓ *	-	-	—	—	—	—	—
Carbohydrate metabolism	Pyruvate	↓ *	↓ *	-	↑ *	↓	↓	↑	↑	-	↓
	Lactate	↓	↓ *	-	-	-	↑	↑	↑	-	↓
	Myo-Inositol	↓	↓ *	-	↑	↓	↑	↑	↑ *	↓	↓
	Glyceric acid	↓	↓	↓ *	↑ *	-	-	↓	↑	-	↑
	Glucose-6-phosphate	↓ *	↑	↓	↓ *	↓	—	—	—	—	—
	Glycerol-3-phosphate	↓ *	↓ *	↓ *	↑ *	-	—	—	—	—	—
	3-Hydroxybutyric acid	↓	↓	-	-	↑	↑	↑	↑	↓	↑ *
	Glucose	-	-	↓	-	↓	-	↑	-	↓	-
	Fructose	↓ *	↓ *	↓ *	↑ *	-	-	↑ *	-	↓	↓
TCA cycle	Succinate	↓	↓	↑	↑	↑	↓ *	↓	↓ *	↑	↓
	Fumarate	↓ *	↓ *	-	-	-	↓	-	↓	-	-
	Malate-3TMS	↓ *	↓ *	↓	↑ *	-	↓	-	↓	↓	↓
	Citrate	↓	↓	-	↑	-	↓ *	↓ *	-	↑	↓
	2-Ketoglutaric acid	↑	↑	↓	↑	↑ *	↓ *	↓	↓ *	↑	↓
	Glutamate-3TMS	↓ *	↓ *	↓ *	↑ *	-	↓	-	↓	↓	↓
	Glutamine-3TMS	↑ *	↑ *	↑ *	↓	↓ *	↑	↑	↑ *	↓	↑
Amino acid metabolism	Aspartate-2TMS	↓ *	↓ *	↓ *	↑ *	-	—	—	—	—	—
	Asparagine	↑ *	↑ *	↑ *	↑	↓	—	—	—	—	—
	Alanine	↑ *	↑ *	↑ *	-	-	↓	↑	↑	↓	↓
	Leucine	↑ *	↑ *	↑ *	-	-	—	—	—	—	—
	Isoleucine	↑ *	↑ *	↑ *	-	-	—	—	—	—	—
	Valine	↑ *	↑ *	↑ *	-	↓	—	—	—	—	—
	5-Hydroxytryptamine	↓ *	↓ *	↓ *	↑ *	-	—	—	—	—	—
	Phenylalanine	↑ *	↑ *	↑ *	-	-	—	—	—	—	—
	Lysine	↑ *	↑ *	↑ *	-	↓	↓	↓	-	-	↓
	Proline	↑ *	↑ *	↑ *	↑	↓	↓	↓	-	-	↑
Glutathione metabolism	Serine-3TMS	↑ *	↑ *	↑ *	-	↓	↑	↑	↑ *	↓	↓
	Cysteine	-	-	-	-	-	↓	↓	-	↑ *	-
	Glycine-TMS	-	↑	↓	↑	↑	↑	↑	↑	-	↓
	Methionine	↑ *	↑ *	↑ *	↑ *	-	-	↑	↑	↓	↓
Purine metabolism	Hypoxanthine	↑	↑	↑	-	↓	↑ *	↑ *	↑	↓ *	↓ *
	Xanthine	↑	↑	↑	-	↓	↑	↑	↑ *	↓	↓ *
	Uric acid	↑	↑	↓	↓	↑ *	↑	↑	↑ *	↓	↓ *
	Ornithine	↑ *	↑	↑ *	↑ *	↓	↓ *	↓ *	-	↑ *	↑
Others	Aminomalonic acid	↑ *	↑	↑	↑	↓	—	—	—	—	—
	Oxalic acid	↑	-	↑	-	-	↑	↑ *	↑ *	↓	↓
	Phosphate	↓ *	↓ *	↓ *	-	-	↓ *	↓	-	↑	↑
	Glutaric acid	—	—	—	—	—	↑ *	↑ *	↑ *	↑	↓
	N-Acetylaspartate-2TMS	—	—	—	—	—	-	↑	↑ *	-	↓
	3-Hydroxyisobutyric acid	—	—	—	—	—	↑	↑	↑	↓	-
	γ-Aminobutyric acid	—	—	—	—	—	↓	↓	-	↑	↓
	Niacinamide	—	—	—	—	—	↓	↑	↑ *	↓	↓
	Hippuric acid	—	—	—	—	—	-	↑ *	↑	↓	↓ *
	2-Hydroxyglutaric acid (3TMS)	—	—	—	—	—	-	↑	-	↑	↑

Note: the sign indicates the direction of change; ↓ * for decreased significantly, ↑ * for increased significantly, - for no change (* *p* < 0.05), ↑ for increase, ↓ for decrease, — for none of the metabolites, as indicated by the statistical analysis using one-factor variance analysis (ANOVA).

## Data Availability

Not applicable.

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
