# Peer review of "Identification of Biomarkers for Methamphetamine Exposure Time Prediction in Mice Using Metabolomics and Machine Learning Approaches"

_metabolites, 2022, doi:10.3390/metabo12121250_

Round 1
Reviewer 1 Report
This paper describes the identification of biomarkers to predict the time of exposure to methamphetamine.The authors validate their results by comparing 6 data processing methods (machine learning).
They also interpret the biochemical disorders caused after administration of methamphetamine and
show that the concentration of biomarkers varies according to the duration of withdrawal.
The results are interesting but the authors describe very little about how the data is processed.
In particular:
- it was necessary to realign the chromatograms?
- Did the authors use a normalization method? They discuss about an internal standard (myristic-1,2-13C2 acid). The authors use a single internal standard?
-
More precisely:
- Explain the abbreviation: TCA
- I don't understand line 42 "where the estimated global number of amphetamines is about 2892,000" = number of amphetamine type?
- Line 169: “strength of each peak”?: the databases give "strength of peak?"
- 3.3 Part: why did the authors compare 6 data processing methods? They do not compare their characteristics. What does one method provide more or less than another?
Author Response
Point 1: it was necessary to realign the chromatograms?
Response 1: Chromatogram alignment is an important procedure for metabolomics data analysis. Time-shift alignment mainly aims to provide comparable information across samples. In our study, we first identified the compound, and got their retention time and m/z; then we integrated the compounds based on this information. We compared all QC samples and found almost no difference in retention times of peaks across different samples (see the figure as following). During integration, we set the time window at 5% of the retention time, and all the peaks were integrated. In conclusion, we performed the alignment during integration by the GCMS solution software.
Point 2: Did the authors use a normalization method? They discuss about an internal standard (myristic-1,2-13C2 acid). The authors use a single internal standard?
Response 2: In this study, we use myristic-1,2-13C2 acid as the internal standard. We also added methyl myristate to monitor the instrument status at the last step of sample preparation. The RSD of methyl myristate was 9.97% and 9.58% in the serum and urine, respectively, and the RSD of myristic-1,2-13C2 acid was 9.58% and 9.76% in the serum and urine, respectively, which indicated that the results were accurate and the method is reliable. By measuring the ratio of the compounds identified and the peak area of myristic-1,2-13C2 acid, the error in the injection process is corrected and eliminated and the relative content of the compounds can be measured.
Point 3: Explain the abbreviation: TCA
Response 3: We have added the explanation of TCA in the manuscript. TCA is the abbreviation of the tricarboxylic acid cycle. Thank you.
Point 4: I don't understand line 42 "where the estimated global number of amphetamines is about 2892,000" = number of amphetamine type?
Response 4: Sorry that we did not describe it clearly. It was the number of amphetamine users. The estimated number of users of amphetamines worldwide is 2,892,000. We have updated the new data in World Drug Report 2021 and added the reference. Thank you.
Point 5: Line 169:”strength of each peak”?: the databases give "strength of peak?"
Response 5: Sorry for the confused expression. In the step of compound identification, we compared the measured spectrum with the spectrum of the reference compound in the metabolite database; we have changed this in the manuscript.
Point 6: 3.3 Part: why did the authors compare 6 data processing methods? They do not compare their characteristics. What does one method provide more or less than another?
Response 6: PCR is an algorithm that solves multivariate collinearity problems through dimensionality reduction (orthogonal transformation), which has the disadvantage of ignoring the correlation between principal components and dependent variables, and PLS is also a multiple linear regression algorithm for problems with small sample sizes but many variables and multiple correlations1. SVM2 is a kernel-based algorithm that projects data into higher-order vector spaces, and performs well in small-sample, nonlinear separable problems, but difficult to train on large-scale samples, sensitive to missing values, and highly demanding on the selection of kernel functions. KNN3 can be easily and intuitively classified and regressed according to the distance of adjacent samples, which has the disadvantage of bias and large calculation for unbalanced data. BRNN4 is a two-way cyclic neural network, and the prediction result is output by considering the previous and last terms of the time series, which has the disadvantage of long running time; RF5 model is an ensemble learning method based on decision trees, which generates multiple trees by random sampling and random sampling of feature sets, summarizes the resulting output of each tree, is not easy to overfit, and is suitable for data with missing data and unbalanced datasets, the established model is stable and has high prediction accuracy, and can be applied to regression and classification problems at the same time, which is one of the best algorithms at present. In our practical application, SVM and BRNN (Supplemental Table 4,5) perform well on the training set, but the performance of the test set is poor, there may be overfitting, and the BRNN model takes a long time to run, thus the random forest is the best model for drug abuse time prediction in comprehensive comparison.
Reference
- Wentzell, P.D.; Montoto, L.V. Comparison of principal components regression and partial least squares regression through generic simulations of complex Chemometrics and Intelligent Laboratory Systems 2003, 65, 257–279.
- Noble, W.S. What is a support vector machine? Nat Biotechnol 2006, 24, 1565-1567.
- Abu Alfeilat, H.A.; Hassanat, A.B.A.; Lasassmeh, O.; Tarawneh, A.S.; Alhasanat, M.B.; Eyal Salman, H.S.; Prasath, V.B.S. Effects of Distance Measure Choice on K-Nearest Neighbor Classifier Performance: A Review. Big Data 2019, 7, 221-248.
- Schmidhuber, J. Deep learning in neural networks: an overview. Neural Netw 2015, 61, 85-117.
- Breiman, L. Random Forests. Mach. Learn. 2001, 45, 5–32.

Reviewer 2 Report
## General
The manuscript with the title "Identification of Biomarkers for Methamphetamine Exposure Time Prediction in Mice using Metabolomics and Machine Learning Approaches" by Sheng et al. investigated the time dependant changes in the metabolome of mice after adminstration of methamphetamine. They compared several statistical methods and suggested biomarker for plasma and urine. This manuscript is well written and scientifically sound. However, I have some small comments which I wish to be addressed before recommending this manuscript for publication.
## Specific comments
Line 40: Why do you refer to the World Drug Report of 2019? There is an up-to-date version from 2021. Please update your references.
Line 114f: Why did you administer methamphetamine using an intra peritoneal injections, instead of oral administration? Nobody injects himself methamphetamine. Please comment.
Line 131: I am afraid, something went wrong with the formatting of your references.
Line 134: I am afraid I do not understand this sentence. Can you please rephrase it?
Line 143: m/z is not a "mass-to-charge ration", since charges can also be negative, but a m/z is never negative. The term "mass-to-charge ration" is deprected. Please refer to the IUPAC definition no. 324 and 356 in DOI: 10.1351/PAC-REC-06-04-06.
Line 149f: Why did you choose these exact models and not for example PC-DFA?
Line 253f: Your results appear logical when you look at the symptoms of an methamphetamine intake. Please discuss them accordingly.
Author Response
Point 1: Why do you refer to the World Drug Report of 2019? There is an up-to-date version from 2021. Please update your references.
Response 1: Thank you for your kind suggestion. We refer to the World Drug Report of 2019 to quote the specific numbers. Now we have added the new data of World Drug Report 2021 and updated the reference.
Point 2: Line 114f: Why did you administer methamphetamine using an intra peritoneal injections, instead of oral administration? Nobody injects himself methamphetamine. Please comment.
Response 2: Thank you for your question. Indeed, some may take drugs with juice or inhale through the nose. However, for drug abusers, stronger stimulation was needed so that a considerable number of abusers choose the injection way, especially drug addicts. Actually, more than 11 million people worldwide consume drugs through injection. That’s why we chose intraperitoneal injection to mimic real drug users.
Point 3: I am afraid, something went wrong with the formatting of your references.
Response 3: Thanks. We have updated the references.
Point 4: Line 134: I am afraid I do not understand this sentence. Can you please rephrase it?
Response 4: Sorry that we did not describe it clearly. We have modified this sentence in the manuscript. As “The supernatant was transferred into a sample bottle and vacuum concentration to dry. Then methoxyamine in pyridine (10 mg/mL) was added for oximeation and trimethylsilanization reagent was added for silanization. Finally, the external standard Methyl myristate in heptane (30 μg/mL) was added. The mixed solution was analyzed using gas chromatography/mass spectrometry (GC/MS).”
Point 5: Line 143: m/z is not a "mass-to-charge ration", since charges can also be negative, but a m/z is never negative. The term "mass-to-charge ration" is deprected. Please refer to the IUPAC definition no. 324 and 356 in DOI: 10.1351/PAC-REC-06-04-06.
Response 5: Thank you for your professional guidance and kind reminder. We have deleted the inappropriate statement in the manuscript.
Point 6: Line 149f: Why did you choose these exact models and not for example PC-DFA?
Response 6: First of all, we hope to use metabolomics technology to find the relationship between the increase in drug use time and metabolism in vivo, which is a regression problem. PC-DFA, PLSDA, SVM, RF, etc. are all commonly used multivariate data analysis methods in metabolomics. But compared to SVM and RF, etc., PC-DFA1 is a simple and powerful classifier that processes noisy data by dimensionality reduction. PCR is an algorithm that solves the multivariate collinearity problem through dimensionality reduction (orthogonal transformation), the disadvantage is that the correlation between the principal component and the dependent variable may be ignored, and PLS can improve this problem, suitable for the problem of small sample size but many variables and multiple correlations2. PCR uses the principal components as the predictor variables for regression analysis, while PC-DFA uses the principal components for classification. In this study, we selected PCR for regresion between matabolomics and drug abuse time. KNN3 can be classified and regressed simply and intuitively according to the distance of adjacent samples, the disadvantage is that it has bias and large calculation for unbalanced data; SVM4 is not affected by sample distribution, is a kernel-based algorithm, projects data to a higher-order vector space, and performs well in small samples and nonlinear separable problems, but is difficult to train on large-scale samples, sensitive to missing values, and has high requirements for the selection of kernel functions. BRNN5 is a two-way cyclic neural network, and the prediction result is output by comprehensively considering the previous and last terms of the time series, which has the disadvantage of long running time; RF6 model is an ensemble learning method based on decision tree, which generates multiple trees by random sampling and random sampling of feature sets, summarizes the result output of each tree, is not easy to overfit, and is suitable for datasets with missing data and unequilibrium. There is no one-size-fits-all approach that is superior in all cases, and we want to select as many models as possible that are fit for purpose and perform well.
Point 7: Line 253f: Your results appear logical when you look at the symptoms of an methamphetamine intake. Please discuss them accordingly
Response 7: Metabolic profiling can identify potential biomarkers for drug addiction, withdrawal, and relapse since drug usage can drastically change metabolism. According to reports, taking heroin caused an increase in energy metabolism and a shift in serotonin levels7. Similarly, another study reported that accelerated tricarboxylic acid activity, increased fatty acid β-oxidation, as well as a noticeable reduction in branched-chain amino acids during methamphetamine administration8. In our study, the body's energy balance was disrupted after successive increasing doses of methamphetamine. The body's demand for energy supply increased, and glucose metabolism was insufficient that pyruvate, lactic acid, glycerol-3-phosphate and fructose were significantly reduced. By increasing lipolysis, fatty acids are released into the blood, β-oxidation and release a large amount of energy for the body. Serum fatty acid content is reduced, i.e. lauric acid, myristic acid, oleic acid and octadecanoic acid, etc. are significantly reduced after the administration of methamphetamine. At the same time, amino acid metabolism is enhanced, and the body begins to consume protein to mobilize amino acid metabolism for energy. Amino acid content increases significantly: phenylalanine, methionine, alanine, proline, serine, valine, leucine, isoleucine, glutamine, aspartic acid were significantly increased, glutamic acid, aspartic acid significantly reduced. TCA cycle is also affected, fumaric acid, malate, citrate, and succinic acid are reduced, while α-ketoglutaric acid is significantly increased. Moreover, most of the fatty acid and glucose metabolism-related substances have a recovery tendency after withdrawal, while metabolites related to amino acid metabolism are difficult to return to the initial level in the short term. Methamphetamine is a powerful CNS stimulant, but at high dosages, it is cytotoxic to human midbrain dopaminergic neurons. A dopamine overflow in the striatum brought on by methamphetamine can result in an excessive glutamate release into the brain9. After administering methamphetamine, we noticed considerable changes in serotonin, glutamate, glutamine, and gamma-aminobutyric acid. These findings were consistent with the hypothesis of the metabolic influence of exposure in addiction and withdrawal10.
Reference
- Gromski, P.S.; Muhamadali, H.; Ellis, D.I.; Xu, Y.; Correa, E.; Turner, M.L.; Goodacre, R. A tutorial review: Metabolomics and partial least squares-discriminant analysis--a marriage of convenience or a shotgun wedding. Anal Chim Acta 2015, 879, 10-23.
- Wentzell, P.D.; Montoto, L.V. Comparison of principal components regression and partial least squares regression through generic simulations of complex Chemometrics and Intelligent Laboratory Systems 2003, 65, 257–279.
- Abu Alfeilat, H.A.; Hassanat, A.B.A.; Lasassmeh, O.; Tarawneh, A.S.; Alhasanat, M.B.; Eyal Salman, H.S.; Prasath, V.B.S. Effects of Distance Measure Choice on K-Nearest Neighbor Classifier Performance: A Review. Big Data 2019, 7, 221-248.
- Noble, W.S. What is a support vector machine? Nat Biotechnol 2006, 24, 1565-1567.
- Schmidhuber, J. Deep learning in neural networks: an overview. Neural Netw 2015, 61, 85-117.
- Breiman, L. Random Forests. Mach. Learn. 2001, 45, 5–32.
- Zheng, T.; Liu, L.; Aa, J.; Wang, G.; Cao, B.; Li, M.; Shi, J.; Wang, X.; Zhao, C.; Gu, R. Metabolic phenotype of rats exposed to heroin and potential markers of heroin abuse. Drug Alcohol Depend. 2013, 127, 177-186.
- Zheng, T.; Liu, L.; Shi, J.; Yu, X.; Xiao, W.; Sun, R.; Zhou, Y.; Aa, J.; Wang, G. The metabolic impact of methamphetamine on the systemic metabolism of rats and potential markers of methamphetamine abuse. Mol. BioSyst. 2014, 10, 1968-1977.
- Ares-Santos, S.; Granado, N.; Moratalla, R. The role of dopamine receptors in the neurotoxicity of methamphetamine. J Intern Med 2013, 273, 437-453.
- Ghanbari, R.; Sumner, S. Using Metabolomics to Investigate Biomarkers of Drug Addiction. Trends Mol. Med. 2018, 24, 197-205.

Round 2
Reviewer 1 Report
The authors responded well to the suggestions.